

# T-complex protein 1 subunit zeta-2 (CCT6B) deficiency induces murine teratospermia

Peiyin Yang[*], Wenjing Tang[*], Huiling Li, Rong Hua, Yan Yuan, Yue Zhang, Yunfei Zhu, Yiqiang Cui and Jiahao Sha

Department of Histology and Embryology, Nanjing Medical University, Nanjing, Jiangsu, China
[*] These authors contributed equally to this work.

## ABSTRACT

**Background**. The CCT complex is an important mediator of microtubule assembly and intracellular protein folding. Owing to its high expression in spermatids, CCT knockdown can disrupt spermatogenesis. In the present report, we therefore evaluated the in vivo functionality of the testis-specific CCT complex component CCT6B using a murine knockout model system.

**Methods**. A CRISPR/Cas9 approach was used to generate $Cct6b^{-/-}$ mice, after which candidate gene expression in these animals was evaluated via qPCR and Western blotting. Testicular and epididymal phenotypes were assessed through histological and immunofluorescent staining assays, while a computer-assisted sperm analyzer was employed to assess semen quality.

**Results**. $Cct6b^{-/-}$ mice were successfully generated, and exhibited no differences in development, fertility, appearance, testis weight, or sperm counts relative to control littermates. In addition, no differences in spermatogenesis were detected when comparing $Cct6b^{+/+}$ and $Cct6b^{-/-}$ testes. However, when progressive motility was analyzed, the ratio of normal sperm was significantly decreased in $Cct6b^{-/-}$ male mice, with nuclear base bending being the primary detected abnormality. In addition, slight decreases in $Cct4$ and $Cct7$ expression were detected.

**Conclusion**. These data indicated that CCT6B is an important regulator of murine spermatogenesis, with the loss of this protein resulting in CCT complex dysfunction, providing a foundation for further studies.

Corresponding authors
Yiqiang Cui, cuiyiqiang@126.com
Jiahao Sha, shajh@njmu.edu.cn

## INTRODUCTION

Spermatogenesis is a complex process in which spermatogonial stem cells differentiate and develop to yield mature spermatozoa in an ordered process (*Hess & Renato de Franca, 2008*), which is composed of three primary steps. First, spermatogonial stem cells proliferate and differentiate to yield primary spermatocytes in what is known as the mitotic amplification phase or the pre-meiotic-phase. The spermatocytes then undergo a second round of meiotic division and develop into round haploid spermatids in the meiotic phase. In the final post-meiotic phase of spermiogenesis, these round spermatids undergo

morphological changes that lead to mature sperm production (*De Kretser et al., 1998*). During this final phase, extensive remodeling occurs including nuclear condensation, acrosome formation, flagellar development, and a loss of most cytoplasmic material (*Dias et al., 2016*).

Sperm motility and structural integrity are closely linked to microtubule assembly and acrosome formation, and the disruption of these processes can have a severe adverse impact on fertility (*Fouquet & Kann, 1994*). Microtubules are composed primarily of tubulin, whereas the acrosome is made of actin. The folding of tubulin and actin is highly conserved in eukaryotic cells, being regulated by the multi-protein chaperonin-containing TCP-1 (CCT) complex (*Dunn, Melville & Frydman, 2001*; *Lopez, Dalton & Frydman, 2015*; *Valpuesta et al., 2002*). This ATP-dependent chaperonin complex assists in the efficient folding of newly generated polypeptides (*Broadley & Hartl, 2009*; *Rothman, 1989*), and is composed of two symmetrical rings containing 8 paralogous subunits encoded by TCP1, CCT2, CCT3, CCT4, CCT5, CCT6A or CCT6B, CCT7, and CCT8 (*Kubota, Hynes & Willison, 1995a*; *Spiess et al., 2004*). The CCT1-8 subunits share a moderate degree of sequence identity ($\sim$30%), and orthologous proteins are highly similar across species, with $\sim$60% identity being shared between mammalian and yeast cells (*Kubota et al., 1994*; *Kubota, Hynes & Willison, 1995b*). The sequence diversity of individual subunits is believed to provide appropriate substrate specificity through interactions with particular protein domains (*Counts, Hester & Rouhana, 2017*). Indeed, the CCT complex assists in the folding of roughly 15% of human proteins (*Thulasiraman, Yang & Frydman, 1999*; *Yam et al., 2008*).

The CCT complex is a key mediator of protein folding in the context of spermatogenesis (*Giuffrida et al., 2006*; *Souès et al., 2003*; *Zhu et al., 2006*). In planarians, all eight CCT complex subunits have been shown to be critical for normal sperm development, and the relative levels of these proteins have the potential to influence the spermatogenic process (*Counts, Hester & Rouhana, 2017*). However, no knockout-based approaches to date have explored the role of CCT in murine spermatogenesis. There are two CCT6 homologs encoded in the human and murine genomes, with CCT6A being ubiquitously expressed whereas CCT6B is restricted to the testes (*Kubota et al., 1997*). The specific expression of CCT6B solely in the testes suggests it may be an important mediator of spermatogenesis, yet no data confirming this hypothesis have been published to date.

Genetic knockout models remain an essential tool for the in vivo assessment of the functions of specific genes, with CRISPR/Cas9 system-based approaches having been used on multiple occasions to target specific loci of interest in mammalian model systems (*Mali, Esvelt & Church, 2013*; *Shen et al., 2013*; *Wang et al., 2013*). Indeed, knockout mice generated via this approach are commonly used for genetic studies (*Castaneda et al., 2017*; *Hua et al., 2019*; *Jiang et al., 2014*; *Zhang et al., 2019*). As such, we herein employed CRISPR/Cas9 approach to generate a 5-bp shift mutation in the fourth exon of the *Cct6b* gene in C57B/6J mice, and we then used these animals to assess the reproductive impacts of CCT6B knockout. The results of these analyses suggest that *Cct6b* is an important regulator of spermiogenesis.

## MATERIALS & METHODS

### Animals

Animal care and treatment protocols were designed based upon the guidelines of the Institutional Animal Care and Use Committee (IACUC) of Nanjing Medical University, and all protocols described herein have received approval from the Animal Ethical and Welfare Committee (Approval No. IACUC-2009002-1).

All mice were obtained from and maintained under SPF conditions in the Laboratory Animal Center of Nanjing Medical University. Mice were housed in a climate-controlled facility (20–22 °C, 50–70% humidity, 12 h light/dark cycle) with free food and water access. All mice were treated humanely and all efforts were made to minimize suffering. When appropriate, mice were euthanized via cervical dislocation prior to tissue sample collection. There were no surviving animals at the end of study.

### Antibodies

Rabbit anti-CCT6B (NBP2-92177) was purchased from Novus. Rabbit anti-$\beta$-TUBULIN was purchased from ABways (AB0039). LIN28A (ab46020) and anti-$\gamma$H2AX (ab26350) were purchased from Abcam. Mouse anti-AC-Tubulin (T6793) was purchased from Sigma. Rabbit anti-SOX9 (AB5535) was purchased from Merck.

### CRISPR/Cas9-mediated $Cct6b^{-/-}$ mouse generation

$Cct6b$-knockout mice were generated via a CRISPR/Cas9 approach as detailed previously (*Wang et al., 2020*; *Zhang et al., 2019*; *Zhu et al., 2020*). Briefly, single-guide RNAs (sgRNAs) targeting CCT6B exon 4 were designed with the following sequences: 5′-GACGAAAGTTCATGCTGAACTGG-3′ and 5′-ATGTTCTAGCCACATCCAAGAGG-3′. The Cas9 and sgRNA plasmids were respectively linearized using AgeI and DraI respectively, and were then purified with a MinElute PCR Purification Kit (Qiagen, Duesseldorf, Germany). A MESSAGE mMACHINE T7 Ultra Kit (Ambion, TX, USA) was used to generate the Cas9 mRNA, while a MEGA Shortscript and Clear Kit (Ambion) was used to prepare the purified sgRNA. Wild-type C57BL/6 superovulated females were then mated with C57BL/6 males to generate zygotes for Cas9 mRNA and sgRNA injection.

### T7EI cleavage assay and sequencing

Genomic DNA (gDNA) was extracted from testes using the Universal Genomic DNA kit (CW2298M; CWBIO) and amplified using Phanta Max super-fidelity DNA polymerase (p525; VAZYME) and the primers listed in Table S2. After agarose gel electrophoresis, PCR products were purified using a PCR cleanup kit (AP-PCR-250; Axygen). A T7EI cleavage assay was then used for genotyping as described previously (*Shen et al., 2013*). Briefly, after having been mixed with Buffer 2 (NEB), purified PCR products were denatured and re-annealed using a thermocycler. Then, the PCR products were digested using T7EI (M0302L; NEB) for 25 min at 37 °C and separated on a 2.3% agarose gel.

### Off-target effect assay

Using the open tool CRISPOR (http://crispor.tefor.net/) (*Concordet & Haeussler, 2018*), we predicted potential off-target sites based on the sequence of the target site. Off-target

site selection was performed as described previously (*Niu et al., 2014*). Briefly, the aNy base–Guanosine–Guanosine (NGG) sequence was chosen as the protospacer adjacent motif (PAM), and sites with eight conserved base pairs proximal to the PAM with three or fewer total mismatches were chosen as potential off-target sites. Potential off-target loci were first amplified from gDNA extracted from the testes of founder mice with the primers listed in Table S2. PCR fragments including the off-target loci were then subjected to the T7EI cleavage assay. Sanger sequencing of PCR products with typical T7EI cleavage bands was then performed.

## Genotyping

Edited founders harboring *Cct6b* frameshift mutations were mated with wild type mice for a minimum of three generations to eliminate the potential effects of off-target gene editing. PCR amplification was used to confirm the genotypes of the resultant offspring (primers: forward, 5′- GCATACTTACTACTCGGAGAGCAT -3′; reverse, 5′- CAGAGATAAGAAGGTGGCATTGGA -3′), and Sanger sequencing was additionally conducted, with the results being analyzed via SnapGene (v.3.2.1).

## qPCR

An RNeasy Plus Micro Kit with on-column DNase digestion (Qiagen Ltd., 74034) was used to extract RNA from murine tissues. A heat-sterilized Teflon micropestle containing 350 µl of RLT buffer and 4 µl of $\beta$-mercaptoethanol was used to homogenize samples, after which RNA was isolated based on provided directions, with samples being maintained on ice at all times. Final RNA samples were eluted in 14 µl of RNase-free water, after which 1 µg of total RNA per sample was used to prepare cDNA with a PrimeScript RT reagent Kit (TaKaRa Bio Inc., RR037A). Both oligo(dT) and random primers were used for the reverse transcription. After a 15 min incubation at 37 °C, reverse transcriptase was heat-inactivated for 5 s at 85 °C. The samples were maintained on ice throughout the RNA extraction and reverse transcription. A qPCR instrument (StepOne-Plus, Applied Biosciences) was then used to conduct triplicate gene expression analyses via a SYBR green approach. All qPCR primers are compiled in Table S1, and 18S rRNA served as a normalization control.

## Western blotting

Western blotting was conducted using a slightly modified version of a previously published protocol (*Zheng et al., 2015*). Briefly, a lysis buffer (7M urea, 2M thiourea, 2% (w/v) DTT) containing a 1% (v/w) protease inhibitor mixture (Pierce Biotechnology) was used to extract proteins, which were subsequently separated via SDS-PAGE and transferred to PVDF membranes. These blots were then blocked for 2 h at room temperature with 5% nonfat milk in TBST, followed by overnight (>12 h) incubation with primary antibodies at 4 °C. Anti-CCT6B was used at a 1:1000 dilution, while anti-$\beta$-TUBULIN was used at a 1:3000 dilution. After three washes with TBST, blots were then probed for 2 h with secondary antibodies, after which the SuperSignalWest Femto Chemiluminescent Substrate Western Blotting detection system (Thermo Scientific) was used to detect protein bands.
## Histological analyses

Murine testes or epididymal tissues were collected from a minimum of three mice per genotype and were fixed for 24 h in modified Davidson's fluid prior to storage in 70% ethanol. A graded ethanol series was then used to dehydrate these samples, which were paraffin-embedded and cut to prepare 5 μm-thick sections that were mounted onto glass slides. Following deparaffinization, these sections were subjected to Periodic Acid Schiff or Hematoxylin and eosin (H&E) staining to facilitate histological analyses.

## Sperm analysis

Sperm analyses were conducted as in prior reports (*Castaneda et al., 2017*; *Jiang et al., 2014*). Briefly, distal and clamping of the cauda region of the right epididymis was conducted for each mouse, after which this section was excised, washed with warm PBS, and added to an Eppendorf tube containing fresh human tubal fluid (HTF) media (Millipore) containing 10% FBS at 37 °C. Clamping was then reversed, and the cauda was pierced with the tip of a scalpel to enable the sperm to diffuse into the medium for 5 m in at 37 °C. Sperm were then diluted using additional medium to enable a sperm motility analysis of a 10 μl sperm suspension via computer-assisted semen analysis detection (Hamilton Thorne Research Inc.).

## Immunofluorescence staining

Following deparaffinization, tissue sections were rehydrated, washed thrice with PBS (10 min/wash), and boiled in 10 mM citrate buffer (pH 6.0) in a microwave to facilitate antigen retrieval. A microwave oven for 10 min. For spermatozoa samples, preparation was instead conducted by spreading cells onto microscope slides and allowed to air-dry. These cells were then fixed for 10 min using 1% paraformaldehyde in PBS.

Immunofluorescent staining was conducted by washing samples thrice with PBST (10 min/wash), followed by a 1 h blocking step using 1% BSA. Samples were then incubated overnight (>12 h) with appropriate primary antibodies at 4 °C, followed by incubation for 2 h with secondary antibodies. Anti-LIN28A was used at 1:500, anti-SOX9 at 1:500, anti-$\gamma$H2AX was used at 1:100, and anti-AC-TUBULIN was used at 1:500. Hoechst 33342 was used to counterstain nuclei for 5 min, after which slides were rinsed with PBS and mounted using VectaShield or Immu-Mount. An LSM800 confocal microscope (Carl Zeiss AG) was then used to image the stained slides.

## Transmission electron microscopy

Ultrastructural analyses were performed as in prior reports (*Hua et al., 2019*). Briefly, testes from adult mice were isolated and fixed overnight using 2.5% (v/v) glutaraldehyde in 0.2 M cacodylate buffer (50 mM cacodylate, 50 mM KCl, and 2.5 mM MgCl2, pH 7.2). After subsequent washing in this buffer, the tissues were cut into ∼1 mm$^3$ pieces and submerged in 1% OsO4 in 0.2 M cacodylate buffer for 2 h at 4 °C. Samples were then washed again prior to overnight immersion in 0.5% uranyl acetate. After dehydration with an ethanol gradient, these samples were embedded in resin using the Low Viscosity Embedding Media Spurr's Kit (EMS, 14300). An ultramicrotome was then used to prepare ultrathin sample
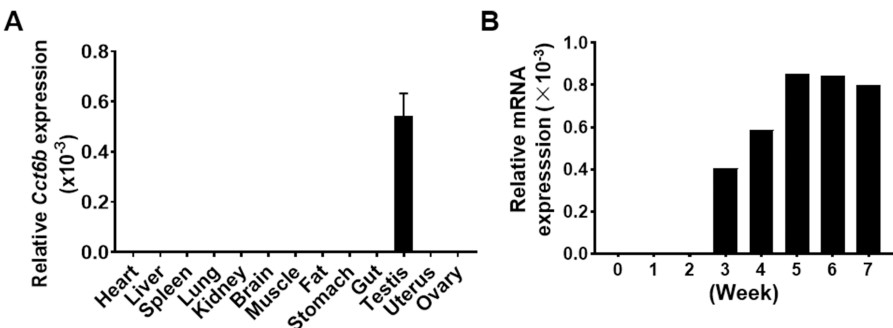

**Figure 1** *Cct6b* **distribution patterns in mice.** (A) The expression of *Cct6b* in different murine tissues (12 weeks) was assessed via qPCR. $n = 3$. (B) The expression of *Cct6b* in murine testes at different developmental stages, including 0, 1, 2, 3, 4, 5, 6, and 7 weeks, was assessed via qPCR. 18s served as a normalization control for all qPCR analyses.

sections that were mounted onto copper grids, stained for 10 min with lead citrate and uranyl acetate, and assessed with a JEM-1400 transmission electron microscope (JEOL).

## Statistical analysis

Data are means ± SEM, and were compared via two-tailed Student's $t$-tests. Not significant (NS): $P \geq 0.05$; *$P < 0.05$; **$P < 0.01$; ***$P < 0.001$; ****$P < 0.0001$.

# RESULTS

## Assessment of murine CCT6B expression patterns

Given that *Cct6b* is evolutionarily conserved (Fig. S1), it is likely to play conserved functions in animals expressing this gene. To explore these functions, we assessed *Cct6b* expression patterns in mice via qPCR, revealing abundant expression of this gene in the testis but not in other analyzed tissues (Fig. 1A). The first round of spermatogenesis in murine testis is relatively synchronous. As such, we collected testis tissue *Cct6b* expression in mice at different numbers of weeks after birth (W) in order to capture the initial spermatogenesis wave. This analysis revealed *Cct6b* expression beginning at week 3, which is the developmental stage during which spermatids emerge in murine testes (Fig. 1B).

## *Cct6b*^−/− mouse generation

To explore the functional importance of *Cct6b* in spermatogenesis, we next used a CRISPR/Cas9 approach to generate C57BL/6 mice in which this gene had been knocked out. To accomplish this, a frameshift mutation in exon 4 of the *Cct6b* gene was introduced into super-ovulated fertilized mouse eggs (Fig. 2A). The mutation causes p.Glu125Gly fs ter22 (Fig. S2), and has the potential to result in the nonsense-mediated decay of the encoded mRNA. Relative to wild-type controls, animals in the F2 *Cct6b*^−/− generation harbored a 5bp deletion in exon 4 of *Cct6b* as detected by PCR and Sanger sequencing (Fig. 2B). Western blotting and qPCR further confirmed that the expression of *Cct6b* was completely absent at the protein level and markedly reduced at the mRNA level in the testes of *Cct6b*^−/− mice (Figs. 2C–2D). Concerns involving off-target mutations associated with

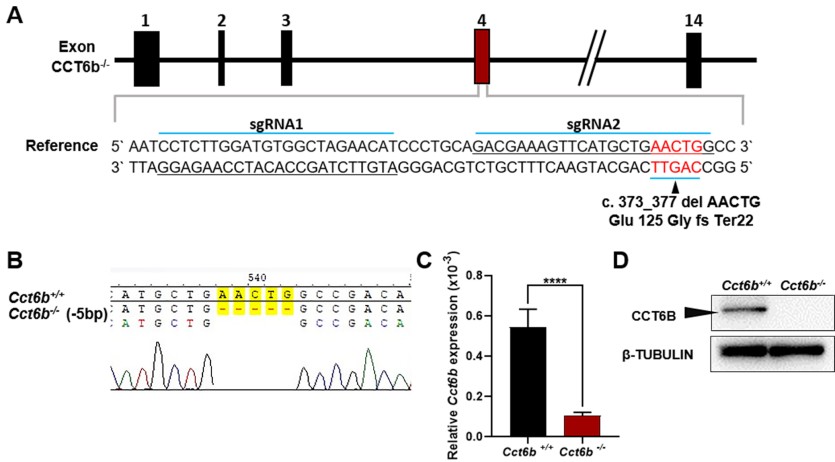

**Figure 2** *Cct6b⁻ᐟ⁻* **mouse generation.** (A) An overview of the CRISPR/Cas9 knockout targeting strategy. (B) Sanger sequencing indicated the presence of a 5-bp deletion in *Cct6b⁻ᐟ⁻* mice. (C) *Cct6b* expression was confirmed to be reduced via qPCR in *Cct6b⁻ᐟ⁻* mice. (D) Western blotting confirmed that testis samples from *Cct6b⁻ᐟ⁻* mice did not exhibit a band of the expected size (58.1 kD). *β*-TUBULIN served as a normalization control.

the CRISPR/Cas9 system have been previously expressed (*Pattanayak et al., 2013*; *Sander & Joung, 2014*), and as such, we analyzed these animals for any potential off-target events. We first excluded the potential off-target effects of the sgRNAs on other CCT genes. Although there is a high degree of homology among the CCT family proteins, there were still multiple mismatched base pairs at the sgRNA site (Figure. S3A). More importantly, protospacer adjacent motif (PAM) sequences at the sgRNA sites of other CCT family members are not conserved, and these play a crucial role in the efficiency of sgRNA cleavage (Fig. S3) (*Cong et al., 2013*; *Mojica et al., 2009*). In addition, predicted off-target sites within the genome were identified, and a subsequent T7E1 assay exhibited no off-target effects on these sites (Table S3 and Fig. S3B). The *Cct6b⁻ᐟ⁻* mice were viable and developmentally normal. Collectively, these results indicated that we had successfully constructed a *Cct6b*-knockout mouse line.

### *Cct6b⁻ᐟ⁻* mice exhibit normal spermatogenesis

The fertility of male mice was next assessed by housing *Cct6b⁺ᐟ⁺* or *Cct6b⁻ᐟ⁻* males with wild-type females for 4 months and recording the number of offspring per litter. The average number of pups per litter for *Cct6b⁺ᐟ⁺* mating pairs was 4.5 ± 1.22, while *Cct6b⁻ᐟ⁻* males sired an average of 5 ± 1.87 pups per litter. This suggests that *Cct6b* is dispensable for male fertility (Fig. 3A). No differences in testes ($n = 3$) or epididymis (Figs. 3B–3C) size were observed when comparing mice in these two groups, and PAS staining did not reveal any differences in spermatogenesis when assessing testes or epididymal tissues from these mice. The average numbers of spermatocytes, round spermatids, and elongated spermatids per tubule were comparable when comparing adult wild-type and *Cct6b⁻ᐟ⁻* testes (Figs. 3D–3E). In addition, immunofluorescent staining revealed normal spermatogonia self-renewal, meiotic progression, and acrosome formation in these two

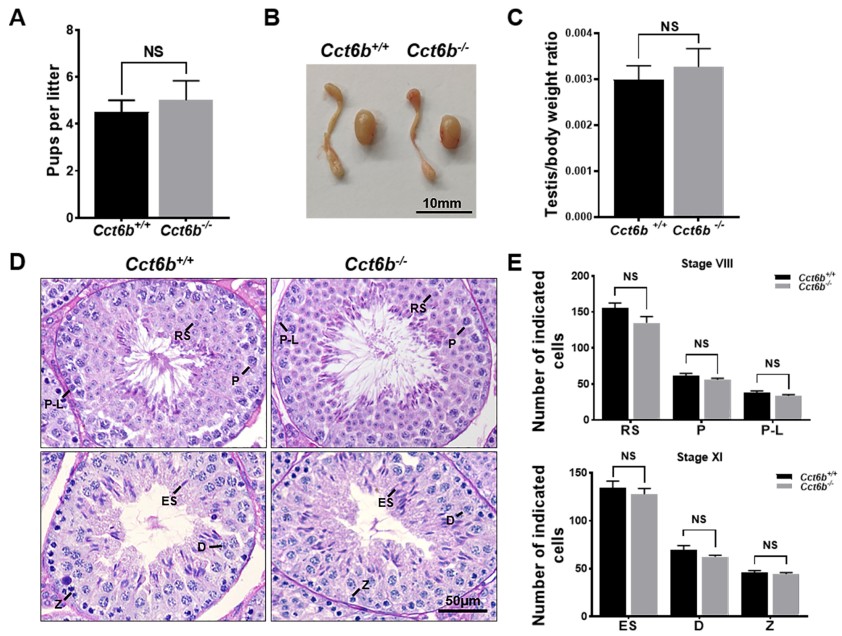

**Figure 3** *Cct6b⁻/⁻* **mice exhibit normal spermatogenesis.** (A) The average number of pups per litter for *Cct6⁺/⁺* and *Cct6b⁻/⁻* mice, *n* = 3. (B) Images of testis and epididymis samples from *Cct6b⁺/⁺* and *Cct6b⁻/⁻* adult mice. (C) Average testis weight normalized to body weight, *n* = 3. (D) Testis sections from *Cct6⁺/⁺* and *Cct6b⁻/⁻* mice following Periodic-acid Schiff staining. (E) The relative composition ratios of different cell types in testis sections at spermatogenic stages VIII and XI. P-L, pre-leptotene; Z, zygotene; P, pachytene; D, diplotene; RS, round spermatids; ES, elongated spermatids.

groups (Figs. 4A–4C), and comparable SOX9 expression was evident in *Cct6b⁺/⁺* and *Cct6b⁻/⁻* mice (Figs. 4D–4E). Together, these results indicated that *Cct6b⁻/⁻* mice are fertile and exhibit normal spermatogenesis.

## CCT6B knockout results in the bending of the sperm neck

After developing in the testes, spermatozoa enter the epididymis. However, no differences were observed when assessing H&E-stained epididymal samples from these two groups of mice (Fig. 5A). We therefore directly assessed sperm quality in *Cct6b⁻/⁻* males with a Computer-Assisted Sperm Analyzer (CASA). The overall epididymal cauda sperm counts were comparable for *Cct6b⁺/⁺* and *Cct6b⁻/⁻* animals (Fig. 5B), and sperm motility was also similar in these two groups (Fig. 5C). However, sperm from *Cct6b⁻/⁻* mice exhibited significantly decreased progressive motility (Fig. 5D). Morphological analyses also revealed that the ratio of normal sperm (*n* = 3, *P* = 0.006) was significantly decreased in male *Cct6b⁻/⁻* mice, with nuclear base bending being the primary abnormality evident in these mice (*n* = 3, *P* = 0.005) (Figs. 5E–5F). However, immunofluorescent staining suggested that the major structures of these malformed sperm, including the flagella and acrosome, remained normal (Fig. 5G). To assess whether the occurrence of bent neck *Cct6b⁻/⁻* sperm was the result of defective flagellar organization, we employed TEM to directly evaluate sperm morphology. No significant differences were observed when comparing flagella between *Cct6b⁻/⁻* and *Cct6b⁺/⁺* sperm, but cytoplasmic retention around the head of

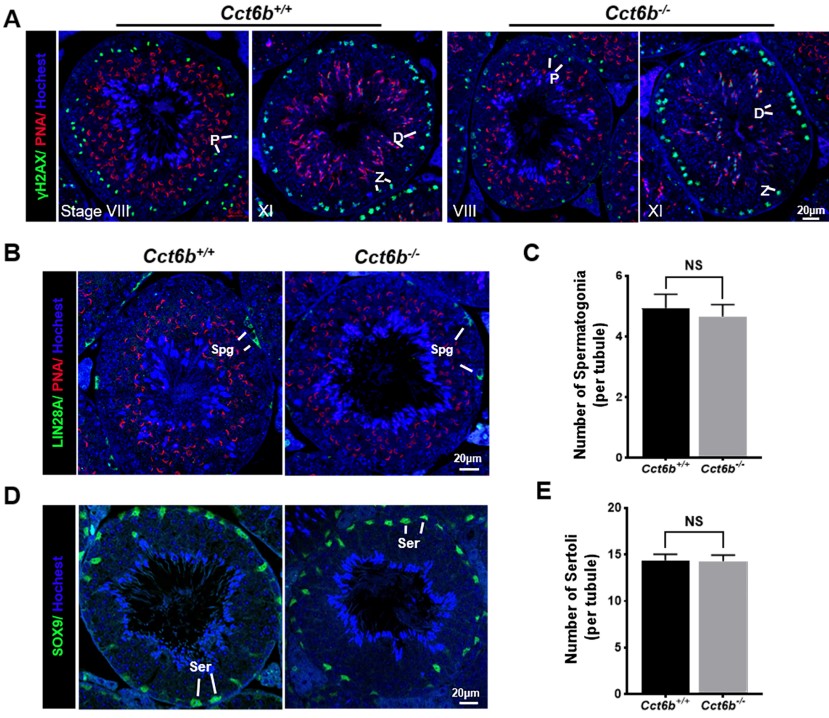

**Figure 4** *Cct6b⁻/⁻* **mice exhibit normal spermatogenic marker expression.** (A) DNA double-strand break formation during meiosis, as measured via γH2AX staining, appeared normal. Spermatids from these knockout mice also exhibited normal acrosomal structures (PNA); Z, zygotene; P, pachytene; D, diplotene; $n = 3$. (B)–(C) Spermatogonia proliferation (LIN28A) did not differ significantly when comparing *CCT6b⁻/⁻* and *CCT6b⁺/⁺* mice; Spg, spermatogonia, $n = 3$. (D)–(E) Sertoli cells (SOX9) numbers in testis sections were comparable in *Cct6⁺/⁺* and *Cct6b⁻/⁻* mice. Ser, Sertoli cell, $n = 3$.

*Cct6b⁻/⁻* sperm was observed via this approach, as was the curling of the flagellar tail in the cytoplasm (Fig. 6). Together, these results indicated that *Cct6b* knockout can result in teratospermia characterized by neck bending and cytoplasmic redundancy.

## Other CCT complex components exhibit decreased transcript abundance in *Cct6b⁻/⁻* mice

Given the function of CCT6B as a CCT complex protein, and the high degree of homology between CCT6B and the other CCT proteins, we additionally explored the impact of knocking out this gene on other CCT complex components at the mRNA level. Compared with the control group, the mRNA expression levels of other CCT family genes were not significantly increased in *Cct6⁻/⁻* testes, (Fig. 7), suggesting that the loss of function of CCT6B was not compensated for by the upregulation of other CCT complex components.

## DISCUSSION

The CCT chaperone complex is estimated to aid in the folding of up to 15% of the eukaryotic proteome (*Thulasiraman, Yang & Frydman, 1999*; *Yam et al., 2008*). Over half of active chaperons are occupied by the folding of tubulin and actin, which serve as

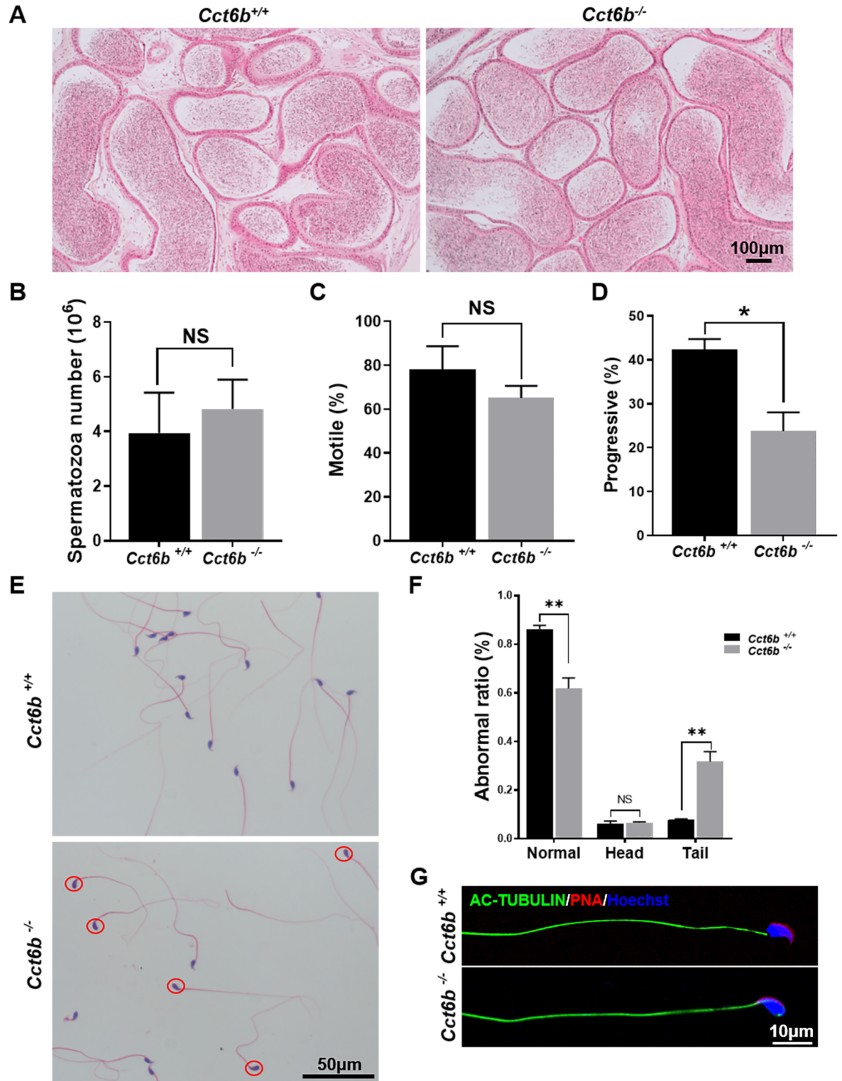

**Figure 5  Teratozoospermia is evident in *Cct6b⁻/⁻* males.** (A) H&E-stained cauda epididymis tissues from *Cct6b⁺/⁺* and *Cct6b⁻/⁻* mice. (B) Cauda epididymal sperm contents from *Cct6⁺/⁺* and *Cct6b⁻/⁻* mice, *n* = 3. (C) Average frequencies of motile and (D) progressive sperm in the *Cct6⁺/⁺* and *Cct6b⁻/⁻* mice, *n* = 3. (E) H&E-stained spermatozoa from *Cct6b⁺/⁺* and *Cct6b⁻/⁻* mice. (F) Frequencies of sperm exhibiting abnormal morphological characteristics in *Cct6b⁺/⁺* and *Cct6b⁻/⁻* mice, *n* = 3. (G) AC-tubulin and PNA immunofluorescent detection in *Cct6b⁺/⁺* and *Cct6b⁻/⁻* spermatozoa.

the primary components of microtubules and the acrosome, respectively, suggesting a potential mechanistic relationship between CCT and spermiogenesis (*Siegers et al., 2008*). Indeed, prior reports have shown CCT proteins to play important roles in spermatogenesis and fertilization. Sylvie et al. initially demonstrated the presence of CCT in the microtubule-organizing centers (MTOCs) and manchette microtubules during spermiogenesis, indicating that this chaperone complex is likely a key regulator of sperm microtubule assembly (*Souès et al., 2003*). Matthew et al. determined that CCT is present on capacitated spermatozoa surfaces and may regulate the zona binding activity of these

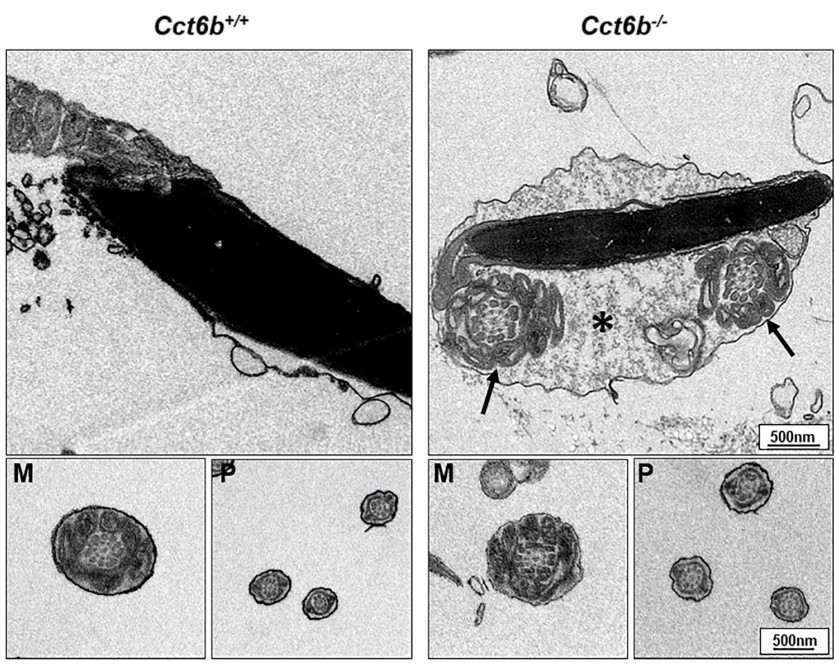

**Figure 6** **Electron microscopy images of *Cct6*⁺/⁺ and *Cct6b*⁻/⁻ spermatozoa.** The cytoplasm of knockout sperms remain around the head of the sperm, while the flagella remain curled in the cytoplasm of these cells (arrows); *, Residual cytoplasm; M, midpiece; P, principal piece.

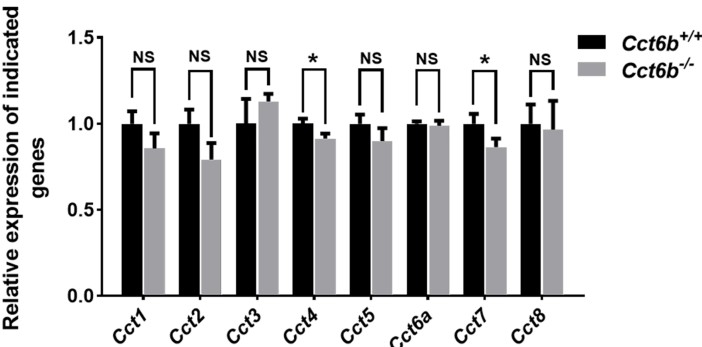

**Figure 7** **Gene expression changes in the testes of *Cct6b*⁻/⁻ mice.** The expression of *Cct1-5*, *Cct6a*, and *Cct7-8* in murine testes was assessed via qPCR. 18S rRNA served as a normalization control. $n = 3$.

cells (*Dun et al., 2011*). CCT knockdown experiments have recently revealed that knocking down any CCT complex subunits can adversely affect fertility in planarians (*Counts, Hester & Rouhana, 2017*). However, no knockout studies in mammalian models have specifically explored the role of CCT in spermatogenesis.

CCT6B is a member of the CCT complex that exhibits a testis-specific expression pattern. Herein, we confirmed that this gene was expressed exclusively during spermiogenesis (Fig. 1), indicating that CCT6B is likely to play an important role in regulating spermiogenesis, spermatid release, and maturation. To test the functional importance of this gene, we

generated *Cct6b*-knockout mice via a CRISPR/Cas9 approach and confirmed that CCT6B protein expression was absent in the testes of these animals (Fig. 2). Adult male *Cct6b*$^{-/-}$ C57BL/6 mice remained fertile, and no differences in the histology or morphology of the testes and epididymides of these mice were observed when compared to wild-type controls (Fig. 3). Spermatogonia proliferation and spermatocyte meiosis also appeared normal in these *Cct6b*$^{-/-}$ animals (Fig. 4), and epididymal sperm numbers and motility remained normal. However, progressive motility and the ratio of normal sperm were significantly decreased in male *Cct6b*$^{-/-}$ mice (Fig. 5). TEM revealed a deficiency in the neck of *Cct6b*$^{-/-}$ sperm (Fig. 6). As CCT paralogs share 65–71% sequence identity across most of their exonic sequences, with the ubiquitously-expressed *Cct6a* protein exhibiting particularly high levels of homology with *Cct6b* (*Counts, Hester & Rouhana, 2017*), we additionally evaluated the expression of the other CCT proteins. This analysis revealed slight reductions in *Cct4* and *Cct7* levels, whereas no changes in other components were detected (Fig. 7), suggesting that no compensatory effects mediated by the expression of other CCT complex components occur in these tissues.

Our results indicate that CCT6B plays an important role in regulating sperm morphogenesis. The loss of CCT6B results in an increased ratio of sperm nuclear base bending, suggesting that the functionality of certain structural proteins in these cells is disrupted. This highlights a potential role for the CCT complex in the assembly of structural proteins during spermatogenesis. Even so, CCT6B-deficient mice generated normal amounts of sperm and remained fertile, suggesting that CCT6B is an auxiliary module in the CCT complex, consistent with its testis-specific expression pattern. One possibility is that it plays an enhanced role in protein folding processes in the context of spermatogenesis, but further research will be required to test this possibility.

## ACKNOWLEDGEMENTS

We thank Jinyang Cai for continuous support with microscopy.

### Funding

This work was supported by the National Key Research and Development Program of China (2018YFC1004201, 2018YFC1003500), National Natural Science Foundation of China (92068109). The funders had no role in study design, data collection and analysis, decision to publish, or preparation of the manuscript.

### Grant Disclosures

The following grant information was disclosed by the authors:
National Key Research and Development Program of China: 2018YFC1004201, 2018YFC1003500.
National Natural Science Foundation of China: 92068109.

## Competing Interests

The authors declare there are no competing interests.

## Author Contributions

- Peiyin Yang, Wenjing Tang and Rong Hua performed the experiments, analyzed the data, prepared figures and/or tables, authored or reviewed drafts of the paper, and approved the final draft.
- Huiling Li, Yunfei Zhu and Yue Zhang performed the experiments, prepared figures and/or tables, and approved the final draft.
- Yan Yuan analyzed the data, authored or reviewed drafts of the paper, and approved the final draft.
- Yiqiang Cui conceived and designed the experiments, performed the experiments, prepared figures and/or tables, authored or reviewed drafts of the paper, and approved the final draft.
- Jiahao Sha conceived and designed the experiments, authored or reviewed drafts of the paper, and approved the final draft.

## Animal Ethics

The following information was supplied relating to ethical approvals (i.e., approving body and any reference numbers):

The Institutional Animal Care and Use Committee (IACUC) of Nanjing Medical University approved this research (No. IACUC-2009002-1).

## Data Availability

The raw measurements are available in the Supplementary Files.

## Supplemental Information

Supplemental information for this article can be found online at http://dx.doi.org/10.7717/peerj.11545#supplemental-information.

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
