# Peer review of "T-complex protein 1 subunit zeta-2 (CCT6B) deficiency induces murine teratospermia"

_PeerJ, doi:10.7717/peerj.11545_

## Round 0.1 · original submission · Major Revisions

Please address all the concerns of the revisers and amend your manuscript accordingly.

Reviewer 1 ·

Basic reporting

The manuscript submitted by Yang et al. show that CCT6B, a testis-specific member of the Chaperonin- Containing Complex, is required for proper development of spermatozoa. More specifically, the authors show that sperm of CCT6B knockout mice have decreased motility and abnormal head morphology. I commend the authors for their extensive data set, compiled over multiple experimental approaches and of detailed analyses. For the most part, the manuscript is clearly written in professional, unambiguous language. However, there are a few items that should be attended to improve the manuscript before Acceptance.

Main edits:

The resulting cct6b gene mutation generated by crispr is not clearly defined. The authors mention that "an in-frame deletion in exon 4 of the Cctb6 gene" was generated, but an in-frame deletion would not result in the absence of protein and transcript described later in the text. Please revise this statement if you meant "out of frame" and provide a figure (main or supplementary) that describes the resultant mutant sequence and predicted mutant protein product generated by this mutation. Most likely, since transcript levels decrease, this mutation generates a premature stop codon that make the transcript suitable for non-sense mediated decay. Please clarify.

Lines 92-95: Please include dilutions and incubation times used for each antibody.

Lines 121-127: Please specify whether the reverse transcription reaction was performed using oligo(dT), random primers, or both. This is relevant because 18S rRNA (which is not polyadenylated) is being used as normalization control.


Minor edits:
Line 54: I believe TCP1 should be included in the following sentence: “The CCT2-8 subunits share a moderate degree of sequence identity (~30%)...”

Line 121: Regarding samples being “maintained on ice at all times”, please clarify whether this is during or after the RNA extraction, or both.

Line 122: “RN” should be “RNA”

Line 161: There is a typo on “antigen retrieval. a microwave oven for 10 min”

Line 194: What is meant by “SA”?

Line 210: Typo in "we", do you mean "was"?

Line 246: "Downregulation" of other cct genes should be edited to "decreased transcript abundance" because it is not clear whether this is due to a physiological side-effect or a direct participation in transcriptional regulatory networks. In addition, it should be specified that this is only about a 10% decrease.

Line 253: Define what is meant by CCR?

Line 273: When mentioning "material", do you mean "motility"?

Cct6b is written as Cctb6 in multiple occasions. Please revise.

Experimental design

no further comment

Validity of the findings

no further comment

Reviewer 2 ·

Basic reporting

No comment

Experimental design

See general comments

Validity of the findings

See general comments

Additional comments

The authors have evaluated the knockdown of CCT6B, which is involved in microtubule assembly, showing an impairment in mice spermatogenesis, namely in progressive motility. It is a very interesting paper, but I have some concerns about the specificity of knockdown using Cas-9. There is a high homology between CCT6B and the others CCTs, and probably targeting the exon 4 of CCT6B may also affect the other CCTs. If exon 4 sequence is CCT6B-specific I suggest to carry out an alignment of sequences demonstrating the specificity. This alignment can be inserted as supplementary figure. In fig. 8, it is speculative to state that downregulation of CCT6B induces downregulation of CCT4 and CCT7 because the possible unspecificity of knockdown and several times the protein expression may not accompaign the mRNA expression.
Also, it is not clear the knockdown of CCT6B. For example, in fig. 2D appears a band with high intensity. Which protein correspond this band? The antibody is unspecific?
The expression of CCT6B should be determined by western blot and not only by qPCR as shown in Fig. 1B.
The article is well writing but there are some mistakes throughout the text that should be reviewed. For example:
- Line 122: it misses “A” in “total RN”
- Line 194: “SA” such

Reviewer 3 ·

Basic reporting

The manuscript is well structured and organized, the figures are relevant, and all the most relevant information presented with sufficient detail in figures and tables (also supplemental). The results presented are relevant to the hypothesis raised.

Authors must improve figure legends as further information is necessary to interpret per see with no need to read the full respective text in the results sections. Namely:
Legend figure 1A - age of animals specifically in this part of the figure
Legend figure 2D - everybody knows what GAPDH is, but a clear reference should still be made. Also, a comment on what is the band above CCT6B.
Legend figure 3 - clarify what stages VIII and XI are.
Legend figure 4 - provide n for this experiments.
Legend figure 6 - insufficient; insert identification of what is in each image and the conclusions from analysing them.
Legend figure 7 - what is the n? What was used as a control gene?

The manuscript is written clearly and presents no major issues in what concerns the usage of English. Some minor typos must be corrected as follows:
- line 20 (abstract) - "...gene expression win these animals..."
- line 194 (results) - "... synchronous, SA such,..."
- line 237 (results) - "... differences we observed..."

Literature references are sufficient. Raw data is provided.

Experimental design

The research described in the manuscript is relevant and novel. It fits the scope of the journal.

The research question per see is well identified, namely identifying the role of cct6b in microtubule assembly in spermatogenesis, given that it is solely expressed in the testes. The experimental work was designed accordingly to confirm the research hypothesis.

The authors performed the experiments using mostly characterization techniques; all were appropriate to the research question. Nevertheless, some details of the methods are insufficiently described. For instance, no details are provided on the dilutions of the antibodies used, which makes difficult the easy and rapid replication of the western blot and immunohistochemistry experiments.

Validity of the findings

The authors replicated experiments to confirm data described in the literature and referred to in the introduction section—namely data presented in figure 1A. While the authors confirming previously published data is relevant in this case - mostly because the reference was more than 20 years old - it would be sufficient to indicate it in the text or reference the original publication close to the confirmation data.

The results are generally well described and presented, and all necessary raw data provided. Still, there are two major issues with the results that the authors should clarify:
1 - The data presented in figure 1D is not clear to us (even considering the raw data provided). The control protein GAPDH band intensity in Cct6b-/- animals is lower than in the Cct6b+/+; does this mean that the loading is inferior in this lane and explains the absence of the CCT6B band? What is the band above Cct6b? Why is the antibody recognizing this band? What are the remaining bands presented in the raw data blot for CCT6B? How sure are the authors that no CCT6B protein is expressed and hence that the KO experiment was completely successful?
2 - On the data provided in Figure 7 about the decrease in the expression of Cct4 and Cct7 in the absence of Cct6b. The decrease is residual when compared to controls. What is the biological significance of this decrease? Does it translate at the protein level? The authors should discuss/explain this. Furthermore, what is the similarity in the sequence between these Cct4 and 7 genes compared to Cct6b? In the introduction, the authors indicate an overall 30% identity. Is it possible that the 5bp deletion designed for Ccct6b affects the Cct4 and 7 genes?

This and other aspects should be further explained and explored in the discussion section, which is short and misses relevant aspects.

Additional comments

The research described in the manuscript is relevant and novel. The experiments are appropriate for the research hypothesis. Nevertheless, some aspects need clarification from the authors.

---

## Round 0.2 · accepted · Accept

All issues pointed by the reviewers were addressed and the revised version is acceptable now.

Reviewer 1 ·

Basic reporting

The authors have addressed my comments and I find the current version of the manuscript acceptable for publication. Thank you and congratulations on your work.

Experimental design

no comment

Validity of the findings

no comment

Reviewer 3 ·

Basic reporting

The authors addressed the issues in the section "1. Basic reporting" raised by this reviewer to the v0 of the manuscript in the revised (v1) version of the manuscript.

Experimental design

The authors addressed the issues in the section "2. Experimental design" raised by this reviewer to the v0 of the manuscript in the revised (v1) version of the manuscript.

Validity of the findings

The authors addressed the issues in the section "3. Validity of the findings" raised by this reviewer to the v0 of the manuscript in the revised (v1) version of the manuscript.